# Bidirectional microwave-optical transduction based on integration of high-overtone bulk acoustic resonators and photonic circuits

Terence Blésin [1,2], Wil Kao [1,2], Anat Siddharth [1,2], Rui N. Wang [1,2], Alaina Attanasio[3], Hao Tian[3], Sunil A. Bhave [3] ✉ & Tobias J. Kippenberg [1,2] ✉

Coherent interconversion between microwave and optical frequencies can serve as both classical and quantum interfaces for computing, communication, and sensing. Here, we present a compact microwave-optical transducer based on monolithic integration of piezoelectric actuators on silicon nitride photonic circuits. Such an actuator couples microwave signals to a high-overtone bulk acoustic resonator defined by the silica cladding of the optical waveguide core, suspended to enhance electromechanical and optomechanical couplings. At room temperature, this triply resonant piezo-optomechanical transducer achieves an off-chip photon number conversion efficiency of $1.6 \times 10^{-5}$ over a bandwidth of 25 MHz at an input pump power of 21 dBm. The approach is scalable in manufacturing and does not rely on superconducting resonators. As the transduction process is bidirectional, we further demonstrate the synthesis of microwave pulses from a purely optical input. Capable of leveraging multiple acoustic modes for transduction, this platform offers prospects for frequency-multiplexed qubit interconnects and microwave photonics at large.

Modern data centers have seen rapidly increasing traffic, which motivates an overhaul of existing network infrastructures. As optical fibers support nearly lossless transport and high bandwidths, twisted-pair copper deployment has shrunk in favor of optical interconnects. Energy-efficient optical transceivers[1] and optical network architectures[2] are being explored in parallel to accommodate emerging data- and resource-intensive applications. In an analogous fashion, processing quantum information in superconducting circuits and networking via photonic interconnects has been envisioned as an effective strategy to address the scalability challenges in advancing quantum technologies[3–5]. The scheme, featuring the full universal set of microwave quantum gates[6] with vanishing thermal occupancy and loss of the optical channels, calls for the

development of microwave-optical transducers to bridge the two energy scales that differ by more than four orders of magnitude. Aside from facilitating the networking of remote quantum processors, these transducers may enable full optical control and readout of microwave qubits[7,8]. Frequency down-conversion of optical input can generate microwave pulses for driving single-qubit gates and readout resonators, whereas the microwave reflection signal of readout resonators can be imprinted onto the optical output as sidebands via up-conversion. This optical manipulation of qubits addresses the scalability challenge. Specifically, the subsequent replacement of coaxial lines bridging room temperature and cryogenic environments by optical fibers is expected to significantly ease the space and heat load constraints in dilution

[1]Institute of Physics, Swiss Federal Institute of Technology Lausanne (EPFL), CH-1015 Lausanne, Switzerland. [2]Center of Quantum Science and Engineering (EPFL), CH-1015 Lausanne, Switzerland. [3]OxideMEMS Lab, Purdue University, West Lafayette, IN, USA. ✉e-mail: bhave@purdue.edu; tobias.kippenberg@epfl.ch

refrigerators, opening up a path toward upscaling processor units housed in a single fridge.

Efficient frequency conversion requires a nonlinear interaction stronger than the coupling to loss channels. The highest conversion efficiency to date has been achieved using an electro-optomechanical approach[9]. There, a silicon nitride membrane interacts with an optical mode of a free-space Fabry–Pérot cavity via radiation pressure and simultaneously serves as the top plate of a capacitor, parametrically coupling the mechanics to the microwave resonator. By virtue of the high resonator quality factors and pump power handling capability, 47% of input photons can be interconverted between optical and microwave domains[10], approaching the 50% efficiency required for attaining finite quantum capacity[11]. This highly efficient transducer has enabled optical dispersive qubit readout with negligible excess backaction[12]. The low-frequency (MHz) mechanical intermediary nevertheless limits the transduction bandwidth and leads to appreciable added noise even at dilution refrigerator temperature. To address these drawbacks, piezo-optomechanical transducers based on optomechanical crystals (OMC) have been developed[13–23], where a tightly confined high-frequency (GHz) mechanical mode and a co-localized optical mode can interact at a vacuum optomechanical coupling rate $g_0 \sim 2\pi \times 500$ kHz. Other piezo-optomechanical transduction platforms have also been developed[24,25]. The trade-off lies instead in the sophistication required for microwave-phonon wave matching that significantly increases design complexity, as well as photothermal instability that constrains the intra-cavity photon number. An on-chip efficiency of 5% has recently been reported on such a platform[21]. However, the low thermal conductance of these suspended quasi-one-dimensional structures hinders the generation of correlated microwave-optical photon pairs at a practical rate. Additionally, their limited power handling makes it challenging to generate a sufficient amount of microwave power for optical manipulation of qubits.

Cavity electro-optic (EO) modulators constitute a conceptually simpler approach where the Pockels effect directly mediates microwave-optical interaction[26–28]. On-chip realizations employing planar superconducting microwave resonators have benefited from the deep sub-wavelength mode volume of the vacuum electric field, reaching vacuum electro-optic coupling rates $g_0 \sim 2\pi \times 1$ kHz[29–32]. However, the material science of $\chi^{(2)}$ crystals poses additional challenges. Photorefractive effects—observed, for instance, in LiNbO$_3$ thin films[33]—hamper optical power handling, while piezoelectric loss and scattered optical photons degrade the quality factor of superconducting resonators[34,35]. The difficulty of producing smooth sidewall surfaces in the workhorse Pockels material, LiNbO$_3$, through dry etching results in propagation losses an order of magnitude above the absorption limit[36]. As a result, the maximum on-chip efficiency achieved with integrated electro-optic transducers exceeds just 2%[29]. In the spirit of ref. 9, a bulk transducer comprising a mm-size mechanically polished LiNbO$_3$ whispering gallery mode (WGM) resonator coupled to a 3D superconducting microwave cavity has proved competitive, trading $g_0$ for improved power handling and quality factors[37,38]. Taking one step further with pulsed optical pumping, the system has demonstrated not only a hallmark 14.4% total efficiency but also electro-optic dynamical backaction[39,40] and microwave-optical quadrature entanglement[41].

With the aforementioned design trade-offs in mind, we present a new integrated piezo-optomechanical transducer based solely on wafer-scale, CMOS-compatible fabrication processes. We utilize a low-loss silicon nitride (Si$_3$N$_4$) photonic molecule, as well as multiple GHz high-overtone bulk acoustic resonances (HBAR) parametrically coupled to the optical modes. The device is endowed with power handling capabilities superior to existing integrated transducers, a feature crucial for the optical manipulation of qubits. However, in contrast to state-of-the-art bulk designs[10,39], it fits compactly within a 100 μm-by-50 μm footprint. In addition, the transduction HBAR

modes are more readily coupled to the microwave signal, unlike OMC transducers. We demonstrate bidirectional microwave-optical transduction with a bandwidth of 25 MHz and total efficiency up to $1.6 \times 10^{-5}$ by pumping with 21 dBm of off-chip optical power in continuous-wave (CW) operation. The device is also characterized by a pulsed optical pump, which constitutes the first step toward the realization of photonic interconnects for qubits and heralded microwave-optical photon pair generation. The simple design, ease of fabrication, robust operation, and compact form factor anticipate wide applicability in quantum technologies and microwave photonics at large.

## Results
### Physics and design
Figure 1 delineates the principle of operation of the present transducer, while the theoretical formalism is detailed in Appendix A. We illustrate the design strategy that distinguishes this first realization of a piezo-optomechanical transducer, leveraging integrated photonics (Si$_3$N$_4$ photonic circuits) and phononics (HBAR) from contemporary integrated OMC and EO transduction schemes. Notably, we design for power handling. For synthesizing the required microwave power for qubit manipulation, a general (second-order) microwave-optical transduction process involves an optical input to be down-converted into a microwave and an optical pump in accordance with energy conservation. The output microwave power is proportional to both the optical pump and optical input power. This underscores the importance of the device's ability to store optical power while avoiding damage or parasitic effects. To this end, we seek to avoid suspended structures with an extremely high aspect ratio, such as OMC, which are susceptible to photothermal instability. Moreover, the commonly employed superconducting microwave resonators are vulnerable to both heating and quasi-particle formation from stray light. We, therefore, opt to pursue a superconductor-less resonator design. Finally, since the Pockels effect transduces light-induced charge fluctuations into the optical domain, Si$_3$N$_4$, a material with vanishing $\chi^{(2)}$-nonlinearity prevalently selected for high-power applications[42], is utilized as optical waveguides. Another practical design consideration is scalability. Towards realizing frequency-multiplexed transduction, a multimode system would be desirable. Due to the disparity in speed between sound and electromagnetic waves, a standing-wave acoustic resonator offers greater compactness compared to its electromagnetic counterpart with the same free spectral range; each of these modes can be independently coupled to optics. Aside from the device footprint, this compactness also leads to significantly reduced parasitic electromagnetic coupling, resulting in improved isolation from the environment. These design constraints motivate the investigation of bulk acoustic waves (BAWs) technology for the microwave subsystem.

The requisite nonlinear interaction for microwave-optical transduction is a parametric three-wave mixing process described by standard optomechanical Hamiltonian

$$\hat{\mathcal{H}}_{\text{int}} = -\hbar g_0 \hat{a}^\dagger \hat{a} \left( \hat{b} + \hat{b}^\dagger \right).$$

Through the bilinear piezoelectric interaction, the acoustic resonance ($\hat{b}$) is coupled to the itinerant transmission line mode ($\hat{c}_{\text{in}}$) of the same frequency $\omega_{\text{in}} = \omega_{\text{m}} \approx 2\pi \times 3.5$ GHz. This frequency matches the detuning of the optical pump from the optical resonance ($\hat{a}$). Specifically, an optical pump addressing the red (blue) side of the resonance induces an effective beam-splitter (two-mode-squeezing) interaction between $\hat{a}$ and $\hat{b}$, as illustrated by the signal flow graph in Fig. 1a and b. The primary figure of merit for a transducer is its efficiency, defined as the number of output photon numbers for each input photon. An efficient device necessitates strong nonlinear interaction between the modes of interest, as well as ease of coupling to these modes internal to the

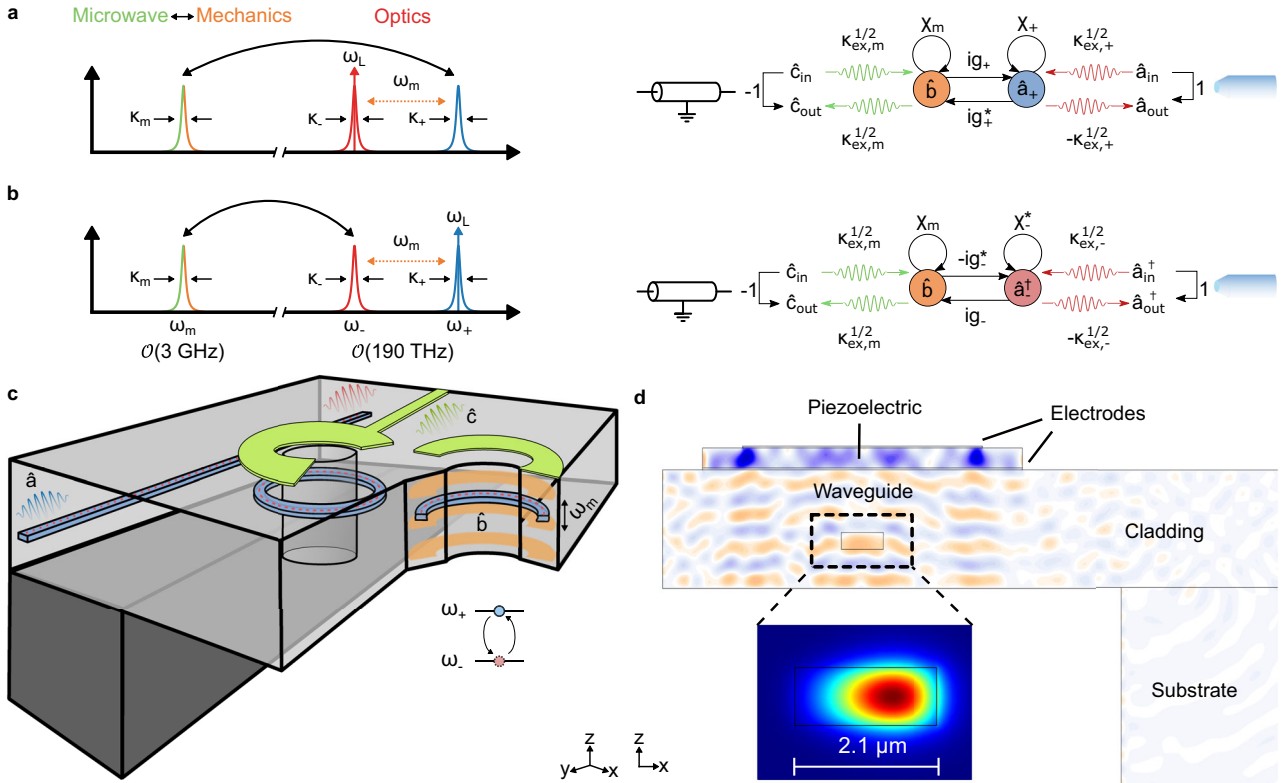

**Fig. 1 | Triply resonant piezo-optomechanical system. a, b** Signal flow graphs illustrating the linearized equations of motion of the system, where the optical pump at frequency $\omega_L$ is set to the anti-Stokes and Stokes configurations, respectively. The optical and acoustic modes with susceptibilities $\chi$ are coupled with (multi-photon) coupling rates $g_\pm = g_{0,\pm}\alpha$, where $\alpha$ is the pump optical field amplitude. Photons are emitted into (out of) the itinerant modes of input optical fiber $\hat{a}_{out}$ ($\hat{a}_{in}$) and microwave transmission line $\hat{c}_{out}$ ($\hat{c}_{in}$) at rates $\kappa_{ex}$. The acoustic and optical modes employed for transduction based on cavity-enhanced three-wave mixing are shown in the frequency domain on the left. **c** Schematic realization of

the transducer (not to scale). A piezoelectric actuator (green) is integrated atop the photonic circuits (blue), allowing the acoustic modes (orange) supported by the suspended cladding (light gray) to be coupled to the microwave transmission line. The optical and microwave ports are labeled by colored wavelets. The inset depicts the energy levels of a photonic molecule that, together with the acoustic mode, fulfills the triply resonant condition. **d** Stack composition and finite-element simulation of the transduction process. Both the mechanical stress pattern of the acoustic mode and the optical TE mode of the micro-ring are shown in the cross-section.

device. Our design addresses these two aspects. First, the internal efficiency

$$\eta_\pm^{int} = \frac{4C}{(1 \pm C)^2}, \qquad (1)$$

depends solely on the three-wave mixing cooperativity $C = 4g_0^2 \bar{n}/(\kappa_o \kappa_m)$. Here, $\bar{n}$, $\kappa_o$, and $\kappa_m$ denote the photon number in the optical cavity, and the total optical and acoustic linewidths of the transduction modes respectively. The vacuum optomechanical coupling rate $g_0 = -(\partial\omega_o/\partial x)x_{ZPF}$, a product of the optical cavity frequency-pull parameter and zero-point displacement of the acoustic wave. In Eq. (1), the plus and minus signs in the denominator correspond to the scenario where the pump is red- and blue-detuned ("anti-Stokes" and "Stokes" processes), respectively[43–45]. Using a triply resonant configuration as in refs. 29–32 enhances the intracavity photon number $\bar{n}$ for a given on-chip input pump power $\eta^{fiber-chip}P_{in}$, where $\eta^{fiber-chip}$ denotes the fiber–chip coupling efficiency and $P_{in}$ the power in the optical fiber. The corresponding photon flux is $\dot{n}_{in} = \eta^{fiber-chip}P_{in}/(\hbar\omega_L)$. The improved photon pumping efficiency can be explicitly seen by considering a "hot" cavity with coupling rate $\kappa_{ex,o}$ driven at a detuning $\Delta = \pm\omega_m$, yielding

$$\frac{\bar{n}}{\dot{n}_{in}} = \frac{\kappa_{ex,o}}{\kappa_o^2/4 + \Delta^2}. \qquad$$

Introducing an additional cavity mode centered at the pump frequency $\omega_L$ makes $\Delta = 0$, leading to an $\bar{n}$ enhanced by $\mathcal{O}(\omega_m^2/\kappa_o^2) = \mathcal{O}(10^3)$ and a significantly reduced laser power overhead required for a given $\eta^{int}$.

The on-chip conversion efficiency $\eta^{oc}$ is related to $\eta^{int}$ by a proportionality constant

$$\eta^{ext} = \frac{\kappa_{ex,o}}{\kappa_o}\frac{\kappa_{ex,m}}{\kappa_m} \qquad (2)$$

such that $\eta^{oc} = \eta^{ext}\eta^{int}$. This extraction efficiency denotes the fraction by which optical and microwave photons are emitted out of the resonator modes with rates $\kappa_{ex,o}$ and $\kappa_{ex,m}$.

Our strategy of optimizing $\eta^{ext}$ is most directly seen in the device implementation, illustrated in Fig. 1d and proposed in ref. 46. The vertical stack comprises a piezoelectric actuator, a $Si_3N_4$ photonic waveguide layer, and a silica ($SiO_2$) cladding that also serves as the acoustic cavity[47,48]. The actuator, deposited on a standard silicon wafer after fabrication of the optical circuitry using the photonic Damascene process, is composed of a c-cut aluminum nitride (AlN) thin film sandwiched between top and bottom molybdenum (Mo) electrodes on which the microwave signal is applied. The alternating electric field launches longitudinal bulk acoustic waves into the cladding layer, which is suspended by etching away the silicon substrate to confine HBARs. Since the AlN is only located above the suspended cladding, the HBAR modes do not extend into the Si substrate

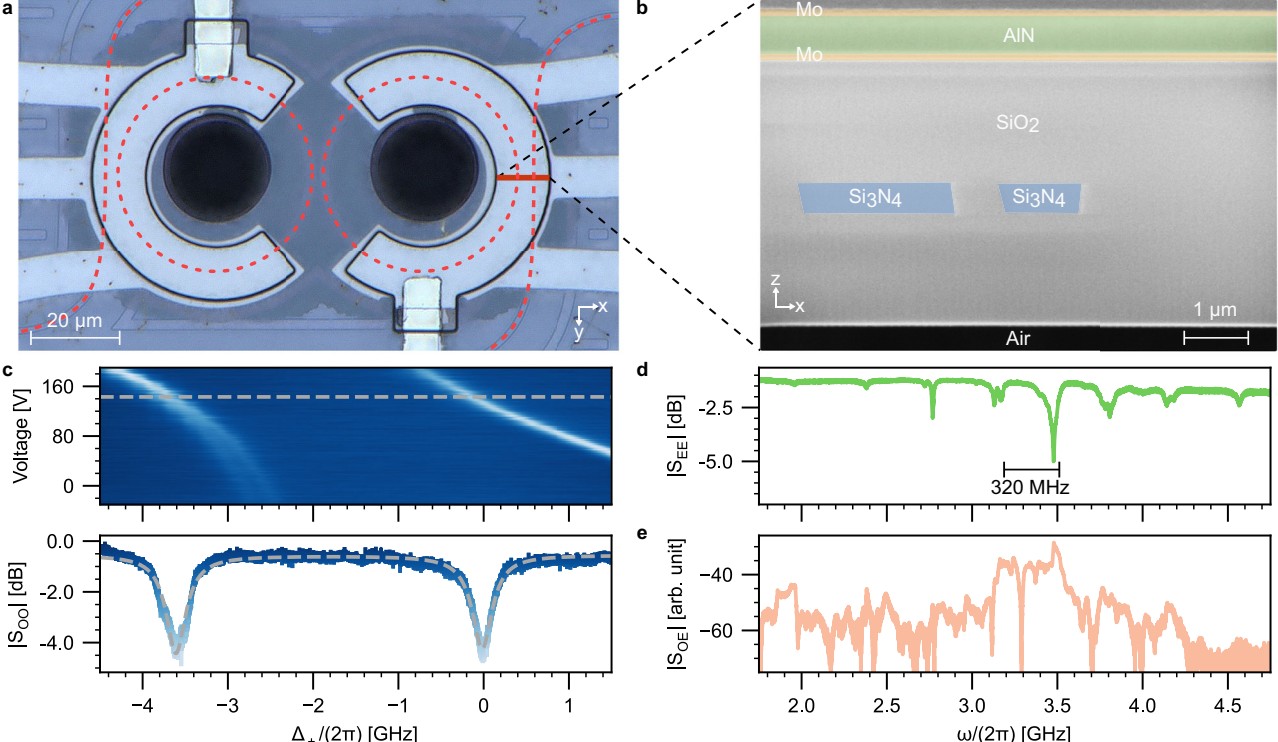

**Fig. 2 | Device metrological and network characterization. a** Optical micrograph of the transducer. The shaded region originates from the released silicon substrate. An additional actuator has been placed on the second ring cavity for DC stress-optic tuning of the optical resonances. The photonic circuits are labeled with red dashed lines. **b** False-color cross-section of the device imaged by focused ion beam scanning electron microscopy showing the top and bottom electrodes (Mo; yellow), piezo-electric layer (AlN; green), suspended cladding (SiO$_2$; gray), and optical waveguides (Si$_3$N$_4$; blue). **c** Optical transmission spectra near 1550 nm for 1 mW optical probe as a function of applied DC piezoelectric control voltage and detuning from the antisymmetric supermode. The transmission color map follows the representative spectrum in the bottom panel. The dashed line denotes the best-fit transmission from coupled mode theory, corresponding to $\kappa_r = 2\pi \times 154$ MHz and $\kappa_l = 2\pi \times 190$ MHz with $\kappa_{ex,r} < 2\pi \times 10$ MHz and $\kappa_{ex,l} = 2\pi \times 120$ MHz in the bare mode picture, and $\kappa_- = 2\pi \times 166$ MHz and $\kappa_+ = 2\pi \times 179$ MHz with $\kappa_{ex,-} = \kappa_{ex,+} = 2\pi \times 60$ MHz in the hybri-dized mode picture (Appendix A1). **d** Microwave reflection spectrum with a probe power of −10 dBm indicating locations of the HBARs. **e** Acousto-optic response spectrum with an off-chip optical pump power $P_{in} = 20$ dBm centered at the sym-metric supermode.

(Appendices B and D). These cavity-enhanced BAWs modify the waveguide refractive index through photoelastic and moving bound-ary effects, realizing the three-wave mixing interaction with rate $g_0$[46]. The stack composition is illustrated in Fig. 1d. Resonances that simultaneously satisfy the interfacial acoustic boundary conditions defined by the actuator and the whole stack naturally feature sizable $\kappa_{ex,m}$. The simplicity of our design is in stark contrast to piezo-optomechanical transducers based on OMCs. To facilitate efficient microwave extraction, design challenges need to be overcome to achieve strong hybridization between the piezo mode supported by the phonon waveguide and the transduction mode of the OMC cavity.

Central to both $\eta^{int}$ and $\eta^{ext}$ are the resonator intrinsic quality factors, as seen in Eqs. (1) and (2). Unlike superconducting microwave resonators, HBARs based on amorphous SiO$_2$ have been shown to exhibit sufficiently low (acoustic) loss at room temperature, relaxing the requirement for cryogenic operation[47]. Two micro-ring resonators are coupled by the overlap of their evanescent fields, leading to the hybridization of the bare cavity modes into symmetric ($\hat{a}_-$) and anti-symmetric ($\hat{a}_+$) supermodes. The resulting photonic molecule makes up the optical portion of the triple resonance system. The waveguides are fabricated using the photonic Damascene process with wafer-scale yield[49,50]. The thick Si$_3$N$_4$ core reduces bending loss, allowing us to employ micro-rings with a radius (free spectral range) of 22 μm (1 THz). Since the hybrid mode splitting $2J$ is proportional to the free spectral range, a gap of 800 nm between the micro-rings is already sufficient for realizing $2J \approx \omega_m$. As illustrated in Figs. 1c and 2a, the micro-ring defines the dimensions of the piezoelectric actuator. Reducing the

micro-ring size, therefore, also results in a reduced acoustic mode volume, which serves to increase $x_{ZPF} \propto 1/\sqrt{V_m}$ and hence $g_0$.

## Device characterization

We perform metrological characterization to affirm the integrity of the stack. Shown in Fig. 2a, the region of suspended cladding is first identified through optical micrography; the released substrate below leads to a contrast in the image. To further access the layer structure, we use focused ion beam milling to create an opening on the device surface at the suspension site. This opening provides sufficient clear-ance to directly image the cladding acoustic resonator, pictured in Fig. 2b, confirming the removal of silicon.

We then characterize the transducer as an optoelectronic network with one microwave port, two optical ports, and one auxiliary electrical DC port. Each optical port comprises an edge-coupled lensed fiber and on-chip bus waveguide terminated by 1D nanotapers designed for TE-polarized light, yielding a fiber-to-fiber coupling efficiency of $\eta^{fiber-fiber} = -8$ dB. For adjusting the frequency splitting between optical supermodes, a bias voltage is applied to the DC port to drive either a piezoelectric actuator[51] (Fig. 2c) or an integrated thermo-optic heater placed in the vicinity of the micro-rings (Figs. 3 and 4; Appendix C2). The former affords no additional heat load and is hence cryogenic-compatible, whereas the latter provides an easy alternative for fast room-temperature characterization. The transduction efficiency depends on the extent to which the triply resonant condition is fulfilled rather than the method by which hybridization is achieved. The avoi-ded mode crossing characteristic of a photonic molecule is exempli-fied in Fig. 2c. We access the HBARs through the microwave port.

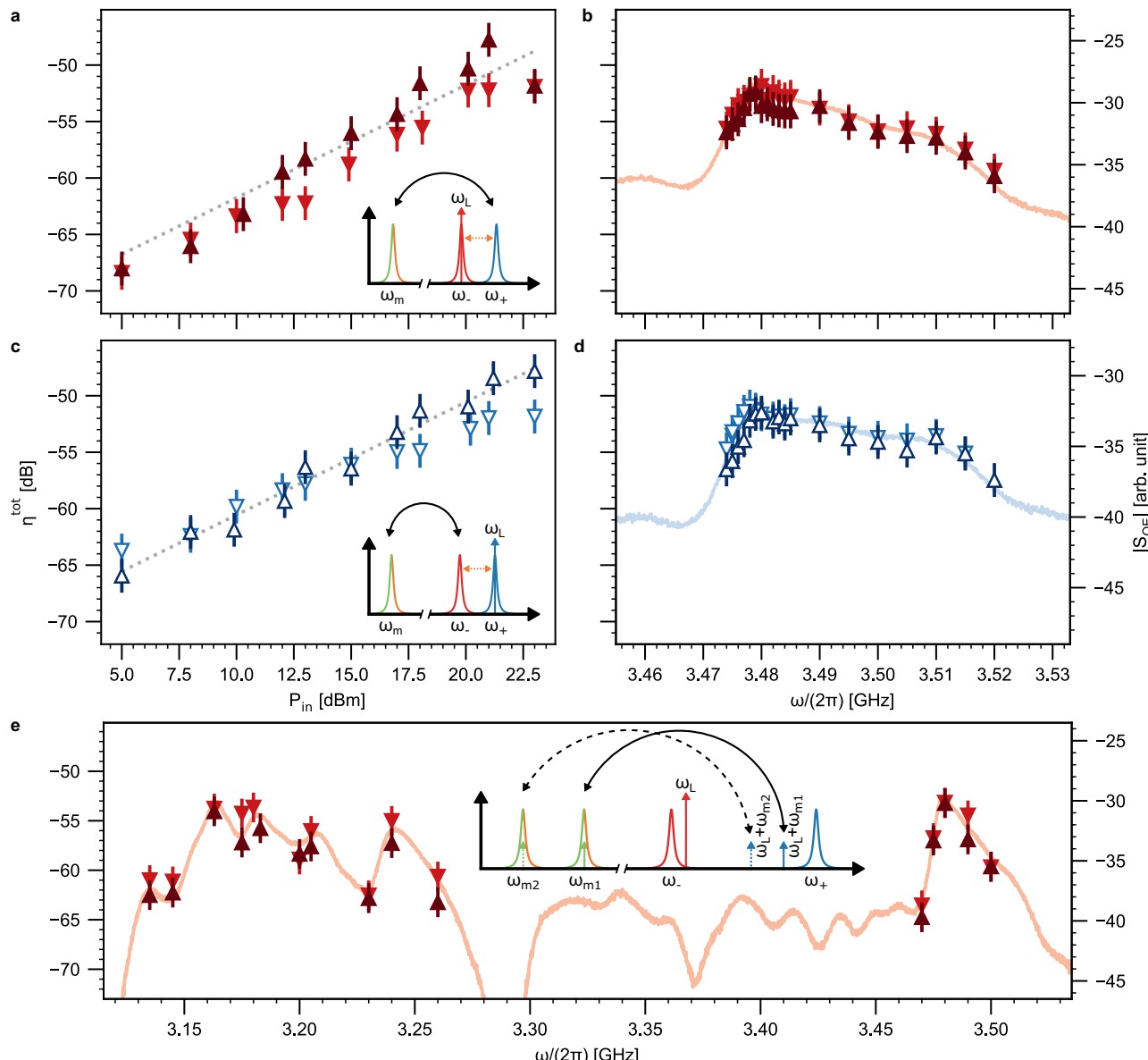

**Fig. 3 | Bidirectional, multimode transduction with a continuous-wave pump.**
**a**, **b** Anti-Stokes transduction. **c**, **d** Stokes transduction. Off-chip photon number transduction efficiencies as a function of off-chip input pump power in the triply resonant configuration are shown in the left panels. Off-chip photon number transduction efficiencies as a function of input microwave frequency and input optical detuning are shown in the right panels, which depict the transduction bandwidth for an input optical pump power of 20 dBm. The up-conversion data were recorded with varying input microwave powers (−20, −10, and 0 dBm) in randomized order. An optical input 20 dB smaller than the optical pump was employed to measure down-conversion data. **e** Bidirectional transduction with a detuned pump leveraging multiple HBAR modes as a function of input microwave frequency and input optical detuning for an input optical pump power of 20 dBm. Up- and down-conversion efficiencies are denoted by triangles and inverted triangles, respectively, with the acoustic-optic response spectrum corresponding to a slight detuning superimposed. The markers are empty for Stokes processes and filled for anti-Stokes processes. The error bars indicate the 95% confidence interval. The dotted lines represent the best-fit efficiency in the low-cooperativity regime where $\eta^{tot} \propto C \propto P_{in}$. The insets illustrate the respective configurations for the pump and input signal. **a** and **c** share the same power axis. **b** and **d** share the same frequency axis. All panels share the same off-chip photon number transduction efficiency axis.

Calibrated high-frequency probes are used to contact the top and bottom electrode pads of the piezoelectric actuator, effectively linking the acoustic resonator with a transmission line. The reflection spectrum (Fig. 2d) reveals predominantly one single series of HBARs with a free spectral range (FSR) of 320 MHz, which corresponds to the acoustic length of the cladding. The transduction acoustic mode at 3.48 GHz exhibits a typical total linewidth of $\kappa_m/(2\pi) = 13$ MHz with a microwave extraction efficiency of $\kappa_{ex,m}/\kappa_m \approx 11\%$ (Supplementary Fig. 6a). The microwave response is distinct from that of an identical stack composition with unreleased substrate. There, a periodic

envelope corresponding to the cladding modes that are more strongly coupled to the microwave is superimposed over the full-stack HBAR response (Appendix D and ref. 47). The absence of these full-stack modes provides another piece of evidence of successful cladding suspension. Finally, we study the device's acousto-optic response in the triply resonant configuration, where the transducer effectively operates as a resonant single-sideband modulator. The optical output containing both the pump at the symmetric-mode frequency and the generated sideband at the antisymmetric-mode frequency are mixed by a photodetector. Such a beat-note spectrum shown in Fig. 2e

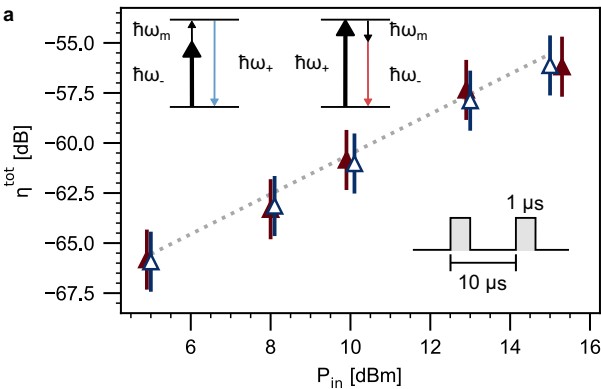

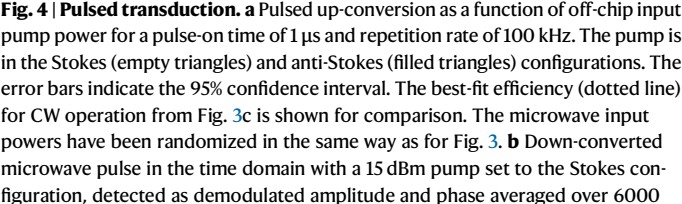

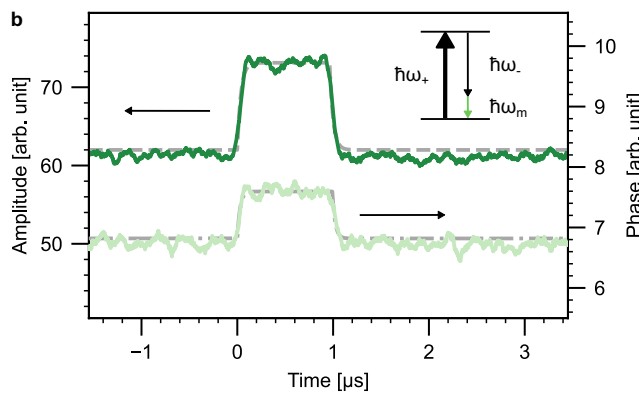

**Fig. 4 | Pulsed transduction. a** Pulsed up-conversion as a function of off-chip input pump power for a pulse-on time of 1 μs and repetition rate of 100 kHz. The pump is in the Stokes (empty triangles) and anti-Stokes (filled triangles) configurations. The error bars indicate the 95% confidence interval. The best-fit efficiency (dotted line) for CW operation from Fig. 3c is shown for comparison. The microwave input powers have been randomized in the same way as for Fig. 3. **b** Down-converted microwave pulse in the time domain with a 15 dBm pump set to the Stokes configuration, detected as demodulated amplitude and phase averaged over 6000 shots. The dashed and dash-dotted lines fit to the RC step response function that physically corresponds to the lock-in integrator, which has a bandwidth (RC time constant $\tau_{RC}$) of 5 MHz (30 ns). The amplitude and phase response yields a best-fit $\tau_{RC} = 35(4)$ and 27(3) ns, respectively. These values are consistent with the lock-in bandwidth. The insets illustrate the respective configurations for the optical pump (thick black), input signal (black), and measured output (colored). The pulse sequence is shown in the time domain.

displays response peaks aligned with the HBAR frequencies in Fig. 2d, demonstrating three-wave mixing.

## Bidirectional microwave-optical transduction

We measure the off-chip photon number transduction efficiency $\eta^{tot}$ of the microwave-to-optical (up-conversion) and optical-to-microwave (down-conversion) processes as a function of the off-chip CW input optical pump power $P_{in}$. Compared to the on-chip efficiency $\eta^{oc}$, this efficiency also accounts for the loss of channels of the microwave and optical ports. First, we study the transducer in the triply resonant configuration. For up-conversion, the optical sideband generated from the microwave input, detuned by $\omega_{in} = \omega_m$ from the pump frequency, is measured via a self-calibrated heterodyne detection method. The off-chip optical output containing both the pump ($\omega_L$) and the sideband ($\omega_s$) is combined with a local oscillator (LO; $\omega_{LO}$) and detected. By placing the LO at a frequency $\omega_{LO} = (\omega_L + \omega_s)/2 + \delta$ such that the photodetector response is constant over a frequency span of $\delta$, we determine the sideband power relative to the pump. Picking off a fraction of the optical output then enables the determination of its absolute power and, by extension, that of the sideband with a power meter. The measured optical sideband power is compared to that of the microwave input to yield $\eta^{tot}$. In the case of down-conversion, the optical input is generated by modulating the phase of the pump with an electro-optic modulator (EOM) driven by a microwave source of frequency $\omega_{EOM} = \omega_m$. Only one of the resulting sidebands is admitted into the photonic molecule and transduced, as the other is far off-resonance. The converted microwave power is directly probed with an electrical spectrum analyzer (ESA). Summarized in Fig. 3a and c, bidirectional transduction processes corresponding to both the effective beam-splitter ($\omega_L = \omega_-$; anti-Stokes) and two-mode-squeezing ($\omega_L = \omega_+$; Stokes) interactions are investigated. Fitting the power dependence of the transduction efficiency yields $g_0 \approx 2\pi \times 42$ Hz (Appendix E). We reach a maximal $\eta^{tot}$ of −48 dB at a 21 dBm pump for each configuration. Knowing the port losses, we estimate an on-chip efficiency $\eta^{oc} = -41$ dB (Appendix E). Accounting for the extraction efficiency $\eta^{ext}$, we further obtain an internal conversion efficiency $\eta^{int} = -27$ dB from Eq. (2).

Furthermore, we deviate from triple resonance to map out the transduction bandwidth. First, for a pump still resonant with one of the optical supermodes, the input microwave frequency $\omega_{in}$ and input optical detuning (controlled by EOM drive frequency $\omega_{EOM}$) are varied for up- and down-conversion, respectively. Shown in Fig. 3b and d,

transduction leveraging the main transduction mode exhibits a full width at half maximum (FWHM) of 25 MHz. The multi-mode nature of the transducer is manifested in Fig. 3e, where the pump is in the beam-splitter configuration but slightly detuned from $\hat{a}_-$. An additional transduction peak in $\eta^{tot}$ with an FWHM of 10 MHz is observed around $\omega_{in} = 2\pi \times 3.165$ GHz, which corresponds to another HBAR one FSR away from the main transduction mode $\omega_m = 2\pi \times 3.480$ GHz. The engineering degrees of freedom, such as cladding and actuator thickness (HBAR FSR) and optical dispersion (supermode splitting as a function of optical wavelength) in the present system, offer possibilities for frequency-multiplexed transduction. The measured device parameters are summarized in Table 1.

## Pulsed transduction

Quantum-enabled operation, where the added noise referred to the transducer input is less than one quanta, would require concurrently attaining high conversion efficiency and low output noise. As the three-

**Table 1 | Summary of the transducer parameters**

| | | |
|---|---|---|
| Optical frequency | $\omega_o/(2\pi)$ | 193 THz |
| Optical free spectral range | $\omega_{o,FSR}/(2\pi)$ | 1 THz |
| Optical inter-resonator coupling rate | $J/(2\pi)$ | 1.77 GHz |
| Fiber-chip coupling efficiency | $\eta^{fiber-chip}$ | −4 dB |
| Symmetric supermode linewidth | $\kappa_-/(2\pi)$ | 166 MHz |
| Anti-symmetric supermode linewidth | $\kappa_+/(2\pi)$ | 179 MHz |
| Symmetric mode extraction efficiency | $\kappa_{ex,-}/\kappa_-$ | 0.36 |
| Anti-symmetric mode extraction efficiency | $\kappa_{ex,+}/\kappa_+$ | 0.34 |
| Microwave frequency | $\omega_m/(2\pi)$ | 3.480 GHz |
| Acoustic free spectral range | $\omega_{m,FSR}/(2\pi)$ | 320 MHz |
| Total acoustic linewidth | $\kappa_m/(2\pi)$ | 13 MHz |
| Microwave extraction efficiency | $\kappa_{ex,m}/\kappa_m$ | 0.11 |
| Vacuum optomechanical coupling rate | $g_0/(2\pi)$ | 42 Hz |
| 3 dB-transduction bandwidth | $B/(2\pi)$ | 25 MHz |
| Total transduction efficiency | $\eta^{tot}$ | −48 dB |
| On-chip transduction efficiency | $\eta^{oc}$ | −41 dB |
| Internal transduction efficiency | $\eta^{int}$ | −27 dB |
| Off-chip optical pump power | $P_{in}$ | 21 dBm |
| Off-chip down-converted microwave power | $P_m$ | −95 dBm |

wave mixing interaction is parametric in nature, the associated cooperativity and, hence, efficiency can, in principle, be enhanced by boosting the optical pump power. A potential trade-off is, nevertheless, the subsequently increased thermal noise. While the high resonance frequency of the transduction HBAR already helps suppress this noise to some extent, additionally employing a pulsed pump presents a number of utilities. Reducing the integrated optical power serves as an effective measure to mitigate thermal load onto the cryostat as well as pump-induced noise while maintaining high peak power. Quantum-enabled operation has, in fact, been achieved with this strategy by a bulk electro-optic transducer employing Watt-scale optical pumping[39]. Gate-based superconducting quantum computers, too, function inherently in the pulsed regime. As such, we characterize our transducer in both frequency and time domains with a pulsed optical pump to evaluate its compatibility with these cryogenic microwave circuits as well as its potential for quantum-enabled operation.

To study pulsed bidirectional transduction, we choose a pulse-on time $\tau_{on} = 1\,\mu s$ and a repetition rate $f_{rep} = 100\,kHz$. We program the pulse sequence such that the increase in temperature, and hence acoustic mode thermal occupancy due to optical heating, is expected to be inconsequential (Appendix C3). First, we measure the up-conversion efficiency, summarized in Fig. 4a, using the same heterodyne method as in the CW case (Fig. 3). The pulsed optical pump mediates transduction of a CW microwave input into a pulsed optical output. This optical pulse comprising both the up-converted output and pump is then down-mixed by the LO, resulting in a microwave pulse whose frequency content can be probed via an ESA. The measured efficiency exhibits reasonable agreement with CW data. Additionally, the down-converted microwave pulse envelope in the time domain provides an independent measure of the transducer bandwidth—a key metric for qubit interconnects[7,8]. With the pulsed pump set in the two-mode squeezing configuration, an optical input pulse is converted on-chip into a microwave pulse with a carrier frequency $\omega_{EOM} = \omega_m$ through difference frequency generation. The time-domain dynamics of the pulse envelope are captured through phase-sensitive demodulation using a digital lock-in amplifier. As shown in Fig. 4b, we observe that the pulse envelope is consistent with the step response of the lock-in integrator, which has a bandwidth of 5 MHz or an RC time constant $\tau_{RC} = 30\,ns$. The lock-in bandwidth, therefore, sets a lower bound for the transducer bandwidth.

## Discussion

Optimization of design and fabrication should engender further improvement in device performance. The conversion efficiency, proportional to microwave extraction efficiency (Eq. (2)), can be improved by doping AlN with scandium[52] to increase its piezoelectric coefficient $d_{33}$ by about five times[53,54] while preserving CMOS compatibility. AlScN provides the additional benefit of increasing the strain induced in the cladding for a given voltage, resulting in a vacuum coupling rate $g_0$ twice higher for high Sc-doping concentrations. Using thinner optical waveguides would reduce acoustic scattering losses, improving the mechanical quality factor as well as the microwave extraction efficiency. There is likewise additional upside on the photonics (Appendix C1). Bending radiation loss can be significantly reduced by increasing the micro-ring radius $r$ or using a material with a higher refractive index. In particular, intrinsic quality factor exceeding $10^7$ has been demonstrated with an $r$ approximately ten times larger using the same fabrication process[55]. As in the thick-core waveguide fabricated using the photonic Damascene process, such a thin-core waveguide on a subtractive manufacturing platform is expected to exhibit negligible bending loss[56]. Moreover, employing higher-index materials such as silicon for the waveguides can offer two benefits. On the one hand, $g_0$ increases with the refractive index. On the other hand, the subsequently improved optical mode confinement enables the use of tighter

waveguide bends and hence reduced acoustic mode volume, further increasing $g_0$. Optimizing the waveguide position inside the cladding and utilizing a higher-order HBAR can each provide an additional twofold improvement in $g_0$. Assuming critical coupling conditions on the optical side (Appendix F), we project another 1.5 times gain in the optical extraction efficiency $\eta^{tot}$. Finally, optical insertion losses can be nearly eliminated by employing a spot-size converter to facilitate mode-matching with lensed fibers[57–59]. Combining these improvements, this transducer could reach $g_0 \approx 2\pi \times 400\,Hz$ and achieve an internal efficiency of 100% for an optical pump power of $\mathcal{O}(10\,dBm)$, limited only by the photon extraction efficiencies and fiber-chip insertion loss.

Non-classically correlated microwave-optical photon pairs can be generated through spontaneous parametric down-conversion in our transducer. These correlated photon pairs constitute a key ingredient toward entangling distant quantum processors via the Duan–Lukin–Cirac–Zoller protocol[4,45,60]. The current on-chip pair generation rate is estimated to be 1.5 kHz, which is well above the thermal decoherence rate for 3.5 GHz at 10 mK, as explained in Appendix A4. However, losses in the measurement setup specific to such an experiment, as well as the gating of the optical pump required to alleviate the effective heat load, are expected to further limit the final heralding rate. Quasi-free-standing structures, such as 1D OMCs, do not readily thermalize and are thus more susceptible to heating effects. Even though our transducer also utilizes a suspended acoustic resonator, it may be feasible to operate at a higher duty cycle than what is presented in Fig. 4. It has been shown that a buffer gas environment can facilitate the thermalization of a 1D OMC at the cost of increased damping of the mechanical breathing mode[61]. On the contrary, the HBAR mode of interest here should be relatively insensitive to such viscous damping, as the acoustic wave propagates predominantly along the longitudinal direction inside the cladding. Therefore, a sample cell affixed to the mixing chamber flange and filled with buffer superfluid helium—an inviscid fluid that is both an excellent thermal conductor and electrical insulator—could potentially improve the heralding rate even further without hampering the conversion efficiency[62].

The present design is also suited for tasks beyond quantum state transfer. The transducer can be a key component of photonic interconnects for superconducting qubits[7,8], where laser lights routed through optical fibers are used to encode microwave signals directly inside the dilution refrigerator during qubit readout. It may be challenging for traveling-wave EOMs to attain the requisite half-wave voltage to be competitive in noise performance against conventional all-electrical readout schemes utilizing high-electron-mobility transistor amplifiers[8]. With a compact footprint of 100 μm-by-50 μm and more than 0.1 pW of microwave power generated on-chip, our triply resonant approach affords a path towards a scalable optically controlled cryogenic waveform generator for both qubit readout and control. The multimode nature of the transducer is particularly suited for frequency-multiplexed dispersive readout. Transduction leveraging multiple HBAR modes is already demonstrated in Fig. 3e. One can additionally utilize line-type instead of point-type dimer coupling for the photonic molecule, which gives rise to dispersion in the supermode splitting[63]. In combination with the multitude of acoustic overtones, a single transducer can, therefore, support several spectrally distinguishable triply resonant systems that serve as qubit multiplex channels. Finally, since the transducer requires no superconducting element to function, it may be pertinent for applications in classical microwave photonics.

In conclusion, we have designed, fabricated, and characterized a compact piezo-optomechanical microwave-optical transducer that integrates wafer-scale, CMOS-compatible HBAR and $Si_3N_4$ photonics technologies. Free of any superconducting components, this triply resonant transducer attains a bidirectional off-chip photon number

conversion efficiency of $1.6 \times 10^{-5}$ ($7.9 \times 10^{-5}$ on-chip, $2 \times 10^{-3}$ internal) and a bandwidth of 25 MHz for an input pump power of 21 dBm at room temperature. Proof-of-principle experiments show that these performances remain unchanged for pulsed optical pumping (1 μs pulse width at 100 kHz repetition rate). Multimode transduction leveraging distinct HBAR modes has been demonstrated, and more than 0.1 pW of microwave power has been generated directly on-chip, suggesting prospects for designing frequency-multiplexed photonic interconnects for superconducting qubits. Realistic design and fabrication improvements may allow access to experiments in the quantum regime.

## Methods

### Device dimensions
On the optical side, the waveguide stack comprises an 850-nm-thick $Si_3N_4$ core embedded in a 5.9-μm-thick $SiO_2$ cladding. The photonic molecule is composed of a pair of micro-ring resonators with a bend radius of 22 μm and a waveguide width of 2.1 μm, separated by a gap of 800 nm. The light is coupled into and out of the resonators via straight 1-μm-wide bus waveguides placed 715 nm away from the micro-ring resonator. For the acoustic subsystem, the arc-shaped piezoelectric actuators each contain a 1-μm-thick AlN layer positioned between 95-nm-thick Mo electrodes. The top and bottom electrodes have widths of 12 and 14 μm, respectively, whereas the piezoelectric AlN is 12 μm wide.

### Device fabrication
The $Si_3N_4$ waveguides are fabricated using the photonic Damascene process[49,50]. The Mo–AlN–Mo stack is sputtered and then patterned using chlorine-based reactive ion etching. For each piezoelectric actuator, the bottom Mo electrode is partially etched back using $XeF_2$. This step serves to avoid short-circuiting the actuator electrodes with the feedline fabricated from an evaporated Al film via a lift-off process. Combining the cladding suspension and chip singulation steps, we employ deep reactive ion etching with $C_4F_8$ chemistry to both open holes in the $SiO_2$ cladding at the center of the actuators and produce trenches to define the chip facets. After a second photolithography that protects the chip facets with photoresist while leaving the etched holes exposed, the Si substrate is isotropically etched away with $SF_6$, suspending the cladding. The chips are finally released by grinding the backside of the wafer.

## Data availability
The code and data generated in this study have been deposited in the Zenodo database under the accession code https://zenodo.org/doi/10.5281/zenodo.10965620.

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

## Acknowledgements

The authors would like to thank Amirali Arabmoheghi, Nils Johan Engelsen, Junyin Zhang, and Shuhang Zheng for experimental assistance and fruitful discussions. This work was supported by the Air Force Office of Scientific Research under award no. FA8655-20-1-7009, Swiss National Science Foundation, under grant agreement no. 204927, and the European Research Council (ERC) under the EU H2020 research and innovation program, grant agreement no. 835329 (ExCOM-cCEO). This work was further supported by the United States Air Force Research Laboratory Award no. FA8750-21-2-0500, as well as United States National Science Foundation's International Collaboration Supplements in Quantum Information Science and Engineering Research for RAISE-TAQS Award no. 18-39164. A.S. acknowledges support from the European Space Technology Centre with ESA Contract No. 4000135357/21/NL/GLC/my. Samples were fabricated in the Center of Micro-NanoTechnology (CMi) at EPFL and Birck Nanotechnology Center at Purdue University.

## Author contributions

T.B. designed the device. R.N.W., A.A., and H.T. fabricated the device. T.B., A.S., and W.K. conducted the experiments. T.B and W.K. wrote the paper with input from all authors. T.J.K. and S.A.B. supervised the project.

## Competing interests

The authors declare no competing interests.
