## [Peer Review File · Nature Communications]

Bidirectional microwave-optical transduction based on integration of high-overtone bulk acoustic resonators and photonic circuitsREVIEWER COMMENTS

Reviewer #1 (Remarks to the Author):

The manuscript by Blésin, et al., presents an experimental demonstration of microwave-optical transducer featuring an integrated optomechanical device design with CMOS-compatible fabrication. The design embeds a high-Q Si₃N₄ optical ring resonator in a SiO₂ high-overtone bulk acoustic resonator. To improve the optical pump efficiency, a second optical ring resonator is designed to couple with the first ring resonator to produce a pair of hybridized modes with a frequency splitting of ~3GHz which matches the acoustic resonance. The bulk acoustic mode is linearly coupled to microwave by depositing a piezoelectric AlN layer sandwiched by a pair of electrodes on top of the SiO₂. A microwave probe is used to contact the electrodes to piezoelectrically actuate and read out the acoustic mode. Using this device, the authors were able to demonstrate bi-directional microwave-optical photon conversion at room temperature with an off-chip efficiency of 1.6×10^{-5} , an on-chip efficiency of 7.9×10^{-5} , and an internal efficiency of 2×10^{-3} . Pulsed transduction was also studied although no obvious improvement in efficiency was observed compared to the cw operation.

Microwave-optical transduction is undoubtedly a very active and important field, especially for quantum information science. The realization of a quantum-enabled transducer holds the key for many advanced technologies such as distributed quantum computing and quantum networks. The design of the transducer in this work was proposed by the authors themselves in a previous theoretical study in Ref [42]. Here, they realize the transducer in experiment, which is new and interesting with certain advantages such as the CMOS-compatible fabrication. However, the performance of the device is unfortunately very poor. Several previous demonstrations have already achieved an on-chip photon conversion efficiency on the order of a few percents (10^{-2}) using various types of integrated transducers such as Ref [21, 29, 32]; namely, the transducer efficiency in this work is 2 to 3 orders of magnitude worse than the state of the art. Moreover, it seems to me that the low performance comes from some intrinsic disadvantages of the device design, which I will discuss in more detail below. In other words, I am not convinced that this design is promising or advantageous for efficient microwave-optical transduction. I cannot consider this work as any significant progress in the field given that there are already many demonstrations of optomechanical or electro-optic transducers with much better performance. In terms of experiment, all the measurements and characterizations done in this work are at room temperature and quite standard. Neither new physics was revealed, nor new experimental techniques were developed. Therefore, I don't think this manuscript is suitable for publication in Nature Communications.

For the transducer design in this work, my biggest concern is that the large mismatch between the optical and mechanical modes and their large mode volumes (especially the bulk acoustic mode) may significantly limit the single-photon nonlinearity. Indeed, the measured single-photon coupling rate is only $g_0/2\pi = 42$ Hz. That is several orders of magnitude smaller than that of a typical optomechanical resonator (the largest can reach ~MHz for 1D optomechanical crystal devices) and about one order of magnitude smaller than integrated electro-optic devices (which is hundreds of Hz, examples are Ref [29, 32]). It means that for the same optical and mechanical/microwave mode linewidths, this bulk acoustic transducer design will require ~100 times higher pump power to achieve the same cooperativity compared to an electro-optic device. In this work, the 10^{-5} efficiency already requires a pump power as high as 21dBm at room temperature. For quantum operations at millikelvin temperatures, where the allowed pump power will be much lower (the typical cooling power is about -10dBm at tens of mK in a dilution fridge), I don't see how practical it is for the authors to boost the transducer efficiency to close to unity.

The authors claim that this bulk acoustic transducer design is more robust and has better power handling capabilities. This is probably true compared with other more delicate optomechanical devices. However, the bulk acoustic resonator is after all still suspended; I don't see how it can be better than integrated electro-optic transducers which have no suspended mechanical structures at all. Another selling point highlighted by the authors is that this transducer doesn't rely on superconducting resonators. However, I don't think utilizing superconducting materials in quantum

transducers is a problem or bottleneck at all because microwave applications have to be done at millikelvin temperatures anyway. Right now, the more urgent tasks are to improve the transducer efficiency and reduce the added noise. In my opinion, sacrificing the transducer performance to eliminate superconducting materials for easier fabrication or larger scale production is a step back rather than an advancement in the field.

In addition, I have some minor technical comments:

1. It would be helpful to provide all the key device parameters in the main text, such as the bulk acoustic resonator thickness, optical and acoustic Q factors, cooperativity, intra-cavity pump photon number, etc. Some of these are only shown in the supplementary material or captions of figures.
2. It took me some time to understand the device structure. The cartoon in Fig. 1(c) and device photos in Fig. 2(a, b) don't really match. Maybe it will help if the authors can indicate the optical rings and coupling waveguide using some dashed lines on top of the Fig. 2(a) photo?
3. It seems that the acoustic Q factor at room temperature is only about 267. What's the limiting factor? Is it limited by radiation loss or material loss? How high acoustic Q will the authors expect at low temperatures?
4. Line 136-137, I think there is a typo in the superscript of η . It should be "fiber-fiber" instead of "fiber-chip" since the text says it's fiber-to-fiber coupling efficiency?

Reviewer #2 (Remarks to the Author):

The paper presents and characterizes a novel design for a bidirectional microwave-optical converter based on a piezo-optomechanical platform. The performance of the device is comparable to that of other platforms which recently appeared in the literature. The idea of using piezoelectrical materials is not novel in the literature, and recent experiments have shown the associated potential. Nonetheless the present paper has the merit to present a novel 2D design which is able to integrate wafer-scale, CMOS-compatible HBAR and Si₃N₄ photonics technologies. A further novelty of the system is its operation in a triply resonant condition, where the mechanical mode is resonant with the splitting between the two coupled optical modes. This allows to operate with very low input power, reduce photothermal effects and open interesting possibilities for quantum operation even at room temperatures.

For this reason I think that the paper is suitable for publication Nature Communications.

However I think that there is room for improvement of the presentation. In fact

1. The author could be more explicitly compare the advantages and the limitation of the proposed device with the converters recently appeared in the literature.
2. For completeness the authors could add the citation to the first theoretical proposals of mechanically mediated optical-microwave converters, which discuss both antiStokes (Y-D. Wang and A. A. Clerk, Phys. Rev. Lett. 108 153603 (2012); L. Tian, Phys. Rev. Lett. 108 153604 (2012)) and Stokes operation (S. Barzanjeh, M. Abdi, G. J. Milburn, P. Tombesi, and D. Vitali, Phys. Rev. Lett. 109, 130503 (2012).) and showed the possibility to operate these devices also in the quantum regime.

Reviewer #3 (Remarks to the Author):

The authors present an integrated optomechanical circuit that can perform frequency conversion between the microwave and optical domains, with similar efficiency when converting in each direction (i.e., microwave-to-optical and optical-to-microwave). The converter uses a combination of high-overtone bulk acoustic waves and coupled photonic ring resonators for the transduction, which is a novel design that offers several advantages. These include high coupling efficiency between the generated acoustic waves and the ring resonators through the photoelastic effect, and the ability to transmit both the optical pump and generated optical sideband frequency

simultaneously and with high efficiency through the photonic integrated circuit. The primary conversion results shown in Fig. 3 are very impressive as they show that bidirectional conversion can be achieved with nearly the same frequency response in both directions. While the presented efficiency is not currently approaching the best reported to date, the novelty of the device, the bidirectional experimental results, the pulsed conversion results that point to operation at low temperature, and the high input optical power handling with linear behavior (i.e., up to 21 dBm, 126 mW) are all strong contributions. Microwave-to-optical frequency conversion is currently of significant interest to the quantum information science and cavity optomechanics communities and this paper is an important contribution to this quickly evolving field. As a result, I recommend this manuscript for publication. I've suggested a few areas below where the authors should expand explanations to improve the reader's understanding.

- What is the electrical impedance of the high-overtone bulk acoustic resonators (HBAR) and how does it affect the transduction efficiency? The planar area of the HBAR seems small compared to other HBARS that are 50 ohms at 3.5 GHz so it would be interesting to know whether the resonator is overcoupled to the microwave source (undercoupled to a sink) and whether this has a positive or negative impact on the transduction results.

- The modes of the coupled ring resonators shown in Fig. 2c are offset by only 3.5 GHz to match with the HBAR mode at that frequency. How was this small offset achieved? I would think that achieving this offset through lithography in a repeatable way would be very challenging. Is one resonator tuned thermally relative to the other? Are chips selected based on their mode split and then tuned? Please explain in the paper.

- Related to the previous question, piezoelectric tuning of the resonances in the coupled ring resonators is shown in Fig. 2c and thermal tuning is described in the Supplementary Information. It's not clear which method was used for collecting the data in Figs. 3 and 4 and whether one method can achieve better results if low temperature operation is not a concern. Please make this clearer in the body of the paper.

- In the Discussion section, the third paragraph would make more sense as the first paragraph. The third paragraph describes how the transducer could be improved through improved design and fabrication. The first and second paragraphs offer more speculative thoughts on how the transducer might be used for quantum control and networking, which while interesting, would be better after the more substantive future improvements to the transducer.

**Response to referees for the manuscript
“Bidirectional microwave-optical transduction based on
integration of high-overtone bulk acoustic resonators and
photonic circuits”**

We appreciate the careful reading of our manuscript by the reviewers and their constructive comments; their suggestions certainly improve the quality of the manuscript.

We present a point-by-point response to each of the reviewers' comments. The reviewers' original suggestions are in black, our replies in blue, and the action taken for improvement in red.

Reviewer #1:

The manuscript by Blésin, et al., presents an experimental demonstration of microwave-optical transducer featuring an integrated optomechanical device design with CMOS-compatible fabrication. The design embeds a high-Q Si₃N₄ optical ring resonator in a SiO₂ high-overtone bulk acoustic resonator. To improve the optical pump efficiency, a second optical ring resonator is designed to couple with the first ring resonator to produce a pair of hybridized modes with a frequency splitting of ~3GHz which matches the acoustic resonance. The bulk acoustic mode is linearly coupled to microwave by depositing a piezoelectric AlN layer sandwiched by a pair of electrodes on top of the SiO₂. A microwave probe is used to contact the electrodes to piezoelectrically actuate and read out the acoustic mode. Using this device, the authors were able to demonstrate bi-directional microwave-optical photon conversion at room temperature with an off-chip efficiency of 1.6×10^{-5} , an on-chip efficiency of 7.9×10^{-5} , and an internal efficiency of 2×10^{-3} . Pulsed transduction was also studied although no obvious improvement in efficiency was observed compared to the cw operation.

Microwave-optical transduction is undoubtedly a very active and important field, especially for quantum information science. The realization of a quantum-enabled transducer holds the key for many advanced technologies such as distributed quantum computing and quantum networks. The design of the transducer in this work was proposed by the authors themselves in a previous theoretical study in Ref [42]. Here, they realize the transducer in experiment, which is new and interesting with certain advantages such as the CMOS-compatible fabrication. However, the performance of the device is unfortunately very poor. Several previous demonstrations have already achieved an on-chip photon conversion efficiency on the order of a few percents (10^{-2}) using various types of integrated transducers such as Ref [21, 29, 32]; namely, the transducer efficiency in this work is 2 to 3 orders of magnitude worse than the state of the art.

Moreover, it seems to me that the low performance comes from some intrinsic disadvantages of the device design, which I will discuss in more detail below. In other words, I am not convinced that this design is promising or advantageous for efficient microwave-optical transduction. I cannot consider this work as any significant progress in the field given that there are already many demonstrations of optomechanical or electro-optic transducers with much better performance.

First, we thank the reviewer for taking the time to review our manuscript. We would like to take this opportunity to better contextualize the performance of the present transducer and, by extension, the advances achieved.

1. The efficiency cited in most papers is the on-chip efficiency, or sometimes the internal efficiency—dictated by the cooperativity. The cooperativity is relevant for the exploration of dynamical backaction phenomena but is not the only relevant quantity for frequency conversion. The ability to convert photons from the input port to the output port is the relevant quantity, which is why we emphasize the total transduction efficiency in this manuscript. Although a fair comparison of frequency converters should include this total port-to-port efficiency, this quantity is often

absent from publications. We quote all three quantities in our manuscript. The table associated with **Figure 1** of this reply shows the most efficient integrated transducers for which we could find the important parameters for comparison. The highest efficiencies so far have been reached by optomechanical crystals and electro-optic transducers. We would like to remark that the transduction bandwidth is also an important figure of merit. Specifically, the “efficiency-bandwidth product” is related to the rate by which correlated microwave-optical photon pairs can be generated, as will be elaborated on later.

2. We stress that by implementing feasible measures, **the present transducer can attain unity internal efficiency and over 50% on-chip and total efficiency**, exceeding the threshold for a generic bosonic channel to exhibit non-zero quantum capacity. The most significant performance gain lies in the single photon nonlinearity, which could be improved in the following ways:

- **Using a higher-order overtone**: Simply increasing the acoustic resonance frequency can enhance the coupling rate. Moving to higher frequencies, where most superconducting circuits operate, could potentially result in a 2 times gain.
- **Optimizing the waveguide position inside the cladding**: Previous simulations showed that placing the waveguide at a displacement maximum should increase the single photon coupling rate significantly ($g_0 > 2\pi \times 400 \text{ Hz}$, cf. T. Blesin et al.). We believe a factor of at least two in g_0 could be gained there.
- **Using a waveguide material with higher photoelastic coefficients or higher refractive index**: Employing materials that exhibit stronger stress-optic effect than silicon nitride such as silicon or lithium niobate could yield significant improvements in the single photon optomechanical coupling rate. Simulations indicate that replacing silicon nitride with silicon alone could result in a factor of two increase in g_0 thanks to its dependence on the third power of the refractive index. Furthermore, silicon rings with smaller bend radii could be manufactured thanks to the tighter confinement of the optical mode in silicon waveguides. Alternatively, lithium niobate waveguides have shown good quality factors for small bend radii (cf. Z. Li et al.) while providing larger photoelastic coefficients. Hence, another factor-of-two improvement in g_0 is expected with waveguide material change.
- **Doping the piezoelectric thin film**: We are currently investigating scandium-doped aluminum nitride actuators for better electromechanical coupling efficiency. This would simultaneously increase the zero-point fluctuations in strain. Preliminary simulations show that more than a factor of two improvement in g_0 should be achievable for high doping concentrations.

Combining these changes would increase the single photon nonlinearity by one order of magnitude, bringing it to levels comparable to electro-optic transducers. Further changes (such as getting closer to critical coupling, reducing the height of the waveguide and improving the fiber-chip insertion loss) would lead to an **internal efficiency of unity for five times less input pump power than the maximum value employed in this manuscript**. We summarize the resulting improved device performance in the table in **Figure 1**.

T. Blésin, H. Tian, S. A. Bhave, and T. J. Kippenberg, Quantum Coherent Microwave-Optical Transduction Using High-Overtone Bulk Acoustic Resonances, Phys. Rev. A 104, 052601 (2021).

Z. Li et al., High Density Lithium Niobate Photonic Integrated Circuits, Nat Commun 14, 1 (2023).

3. Our current device already attains state-of-the-art performance based on “application-centric” metrics. As none of the integrated transducers has reached the

quantum capacity threshold of > 50% total efficiency (cf. C. L. Rau et al., M. Zhang et al. and C. Zhong et al.), teleportation-based protocols have been proposed to circumvent the efficiency constraint for quantum state transfer. This has motivated a series of recent experiments on microwave-optical pair generation (cf. R. Sahu et al., W. Jiang et al., S. Meesala et al.). In this context, the on-chip heralding rate is a key figure of merit for transducers, which we plot in the y-axis of **Figure 1** of this reply (roughly translated into the transducer “gain-bandwidth product”). The heralding rate serves as an indicator of the potential of the transducer to be used in teleportation protocols. As illustrated therein, the transducer presented in this manuscript is competitive with the state-of-the-art thanks to its large bandwidth.

C. L. Rau, A. Kyle, A. Kwiatkowski, E. Shojaei, J. D. Teufel, K. W. Lehnert, and T. Dennis, Entanglement Thresholds of Doubly Parametric Quantum Transducers, *Phys. Rev. Appl.* 17, 044057 (2022).

M. Zhang, C.-L. Zou, and L. Jiang, Quantum Transduction with Adaptive Control, *Phys. Rev. Lett.* 120, 020502 (2018).

C. Zhong et al., Proposal for Heralded Generation and Detection of Entangled Microwave–Optical-Photon Pairs, *Phys. Rev. Lett.* 124, 010511 (2020).

R. Sahu, L. Qiu, W. Hease, G. Arnold, Y. Minoguchi, P. Rabl, and J. M. Fink, Entangling Microwaves with Light, *Science* 380, 718 (2023).

W. Jiang, F. M. Mayor, S. Malik, R. Van Laer, T. P. McKenna, R. N. Patel, J. D. Witmer, and A. H. Safavi-Naeini, Optically Heralded Microwave Photon Addition, *Nat. Phys.* 1 (2023).

S. Meesala, S. Wood, D. Lake, P. Chiappina, C. Zhong, A. D. Beyer, M. D. Shaw, L. Jiang, and O. Painter, Non-Classical Microwave-Optical Photon Pair Generation with a Chip-Scale Transducer, arXiv:2303.17684.

4. We would like to underscore that the transducer would be an attractive candidate for realizing cryogenic interconnects for optical readout and control of superconducting qubits, which is reflected in the second “application-centric metric” of microwave power generated via frequency down-conversion (x-axis of **Figure 1**). This quantity is given by $P_{\text{mw}} = \eta^{\text{oc}} P_{\text{oc}} \frac{\omega_m}{\omega_o} \propto P_{\text{oc}}^2$. Note that here the on-chip efficiency $\eta^{\text{oc}} \propto P_{\text{oc}}$, where P_{oc} is the on-chip pump power. For simplicity, the input optical power is assumed to be equal to P_{oc} as well. In **Figure 1**, we take the maximum pump power used in each work as P_{oc} . Given that the number of photons in the superconducting readout cavity is in general a few hundreds, the efficiency achievable with the simple improvement outlined in the conclusion of the manuscript would be sufficient for direct optical readout. On the control side, we showed that this transducer already generates more than 0.1 pW of microwave power through down-conversion (the highest measured to the best of our knowledge). This is enough power to synthesize sufficiently short qubit control pulses. By contrast, other integrated transducers such as optomechanical crystals and electro-optic transducers face more engineering challenges sustaining the requisite pump power (thermal and photorefractive effects respectively).
5. Finally, for completeness, we compare our new transduction scheme with existing classes of transducers, which we also discuss quite extensively in the introduction of our manuscript.
 - In terms of efficiency, the transducers based on optomechanical crystals (OMC) stand out. However, the main drawback of this approach is scalability. First, in state-of-the-art OMC transducers, magnetic tuning of the superconducting

microwave resonator is required, and proper magnetic shielding would be critical when interfacing with superconducting qubits. Moreover, the subwavelength optical and mechanical mode volume hampers power-handling capability. Consequently, generating sufficient microwave power through frequency down-conversion may be challenging.

- Excluding the transducers that require magnetic tuning, the only realizations that compete with the present transducer are the electro-optic transducers from Hong Tang's group (Yale; aluminum nitride [1] and lithium niobate [6]) and the latest work from Marko Lončar's group (Harvard; lithium niobate [15]). Please refer to **Figure 1** of this reply and the aforementioned improvement roadmap on how we plan to bridge and surpass this gap.

[1] L. Fan, C.-L. Zou, R. Cheng, X. Guo, X. Han, Z. Gong, S. Wang, and H. X. Tang, Superconducting Cavity Electro-Optics: A Platform for Coherent Photon Conversion between Superconducting and Photonic Circuits, *Science Advances* 4, eaar4994 (2018).

[2] W. Fu et al., Cavity Electro-Optic Circuit for Microwave-to-Optical Conversion in the Quantum Ground State, *Phys. Rev. A* 103, 053504 (2021).

[3] J. Holzgrafe, N. Sinclair, D. Zhu, A. Shams-Ansari, M. Colangelo, Y. Hu, M. Zhang, K. K. Berggren, and M. Lončar, Cavity Electro-Optics in Thin-Film Lithium Niobate for Efficient Microwave-to-Optical Transduction, *Optica* 7, 1714 (2020).

[4] W. Jiang, F. M. Mayor, S. Malik, R. Van Laer, T. P. McKenna, R. N. Patel, J. D. Witmer, and A. H. Safavi-Naeini, Optically Heralded Microwave Photon Addition, *Nat. Phys.* 1 (2023).

[5] T. P. McKenna et al., Cryogenic Microwave-to-Optical Conversion Using a Triply Resonant Lithium-Niobate-on-Sapphire Transducer, *Optica*, OPTICA 7, 1737 (2020).

[6] Y. Xu, A. A. Sayem, L. Fan, C.-L. Zou, S. Wang, R. Cheng, W. Fu, L. Yang, M. Xu, and H. X. Tang, Bidirectional Interconversion of Microwave and Light with Thin-Film Lithium Niobate, *Nat Commun* 12, 4453 (2021).

[7] X. Han et al., Cavity Piezo-Mechanics for Superconducting-Nanophotonic Quantum Interface, *Nat Commun* 11, 1 (2020).

[8] A. Khurana, P. Jiang, and K. C. Balram, Piezo-Optomechanical Signal Transduction Using Lamb-Wave Supermodes in a Suspended GalliumArsenide Photonic-Integrated-Circuit Platform, *Phys. Rev. Appl.* 18, 054030 (2022).

[9] M. Mirhosseini, A. Sipahigil, M. Kalaei, and O. Painter, Superconducting Qubit to Optical Photon Transduction, *Nature* 588, 7839 (2020).

[10] L. Shao et al., Microwave-to-Optical Conversion Using Lithium Niobate Thin-Film Acoustic Resonators, *Optica*, OPTICA 6, 1498 (2019).

[11, 12] M. J. Weaver, P. Duivesteyn, A. C. Bernasconi, S. Scharmer, M. Lemang, T. C. van Thiel, F. Hijazi, B. Hensen, S. Gröblacher, and R. Stockill, An Integrated Microwave-to-Optics Interface for Scalable Quantum Computing, *Nat. Nanotechnol.* 1 (2023).

[13, 14] T. C. van Thiel et al., High-Fidelity Optical Readout of a Superconducting Qubit Using a Scalable Piezo-Optomechanical Transducer, arXiv:2310.06026.

[15] H. K. Warner et al., Coherent Control of a Superconducting Qubit Using Light, arXiv:2310.16155.

An estimation of the g_0 value that can be obtained with simple design changes is now given explicitly in the Discussion section.

ID	Group	Input fiber power [W]	Fiber-chip insertion	On-chip efficiency	Total efficiency	Bandwidth [Hz]
1	Tang	2.5e-02	2.5e-01	2.0e-02	5.0e-03	5.9e+05
2	Tang	1.2e-02	2.5e-01	2.5e-05	6.3e-06	2.9e+06
3	Loncar	1.0e-03	1.0e-01	2.7e-05	2.7e-06	1.3e+07
4	Safavi-Naeini	4.0e-05	2.5e-01	4.9e-02	1.2e-02	1.5e+06
5	Safavi-Naeini	1.2e-03	6.0e-02	6.6e-06	3.9e-07	2.0e+07
6	Tang	3.2e-01	6.3e-02	1.0e-02	6.3e-04	6.0e+06
7	Tang	4.0e-02	2.0e-01	7.3e-04	1.5e-04	1.0e+06
8	Balram	5.0e-03	8.0e-02	1.2e-07	9.6e-09	6.0e+06
9	Painter	2.0e-06	6.5e-01	1.0e-03	8.8e-06	1.5e+06
10	Loncar	1.0e-03	3.2e-01	1.7e-05	5.4e-06	1.0e+06
11	Stockill	1.0e-06	4.2e-01	9.0e-03	3.8e-03	1.5e+07
12	Stockill	1.0e-06	4.2e-01	5.2e-05	2.2e-05	1.5e+07
13	Groeblicher	6.0e-06	2.8e-01	1.1e-02	3.1e-03	5.3e+05
14	Groeblicher	6.0e-06	2.8e-01	8.0e-04	2.2e-04	5.3e+05
15	Loncar	4.2e-04	1.0e-01	1.2e-02	1.2e-03	3.0e+07
16	This work	1.3e-01	4.0e-01	7.9e-05	1.6e-05	2.5e+07
17	Anticipated improvements	1.3e-01	9.0e-01	5.6e-01	5.1e-01	2.5e+07

Figure 1: Comparison of leading integrated microwave-optical transducers.

In terms of experiment, all the measurements and characterizations done in this work are at room temperature and quite standard.

Neither new physics was revealed, nor new experimental techniques were developed. Therefore, I don't think this manuscript is suitable for publication in Nature Communications.

While we have already given a comprehensive response addressing the point of novelty, here we highlight additional aspects that separate our work from other microwave-optical transduction papers:

- So far, integrated transducers have been optimized for one single microwave resonance. Here, we show transduction using multiple mechanical modes without sacrificing the efficiency.
- Operation in a dilution fridge will most likely require the optical pump to be pulsed. However, even efficient transducers such as the one showed by M. Weaver et al. have exhibited reduced efficiency in the pulsed regime. We showed that the transduction process does not degrade even for high repetition rates of short pulses.
- Optically driving single qubit operation will require generating relatively powerful pulses using transducers. To the best of our knowledge, we (explicitly) showed the highest microwave power generated on-chip through down-conversion, which, as previously mentioned, depends on not only transduction efficiency but also optical power handling.
- The implementation of the transducer is quite different from other approaches. Most of other transducers rely on either Pockels materials (mostly lithium niobate) or optomechanical crystals. This is a first demonstration of transduction using bulk acoustic waves (BAW) and photonic waveguides integrated on the same chip. Given the aforementioned roadmap toward unity internal efficiency, our transduction scheme may inspire a series of follow-up designs leveraging BAW on different material platforms.

No action taken.

For the transducer design in this work, my biggest concern is that the large mismatch between the optical and mechanical modes and their large mode volumes (especially the bulk acoustic mode) may significantly limit the single-photon nonlinearity.

Please refer to the above discussion on our strategy to improve the single-photon nonlinearity toward attaining unity internal efficiency.

An estimation of the g_0 value that can be obtained with simple design changes is now given explicitly in the Discussion section.

Indeed, the measured single-photon coupling rate is only $g_0/(2\pi) = 42$ Hz. That is several orders of magnitude smaller than that of a typical optomechanical resonator (the largest can reach \sim MHz for 1D optomechanical crystal devices) and about one order of magnitude smaller than integrated electro-optic devices (which is hundreds of Hz, examples are Ref [29, 32]). It means that for the same optical and mechanical/microwave mode linewidths, this bulk acoustic transducer design will require \sim 100 times higher pump power to achieve the same cooperativity compared to an electro-optic device. In this work, the 10^{-5} efficiency already requires a pump power as high as 21 dBm at room temperature. For quantum operations at millikelvin temperatures, where the allowed pump power will be much lower (the typical cooling power is about -10 dBm at tens of mK in a dilution fridge), I don't see how practical it is for the authors to boost the transducer efficiency to close to unity.

After implementing the steps described in this reply as well as the Discussion section of the main text, a **unity internal efficiency** could be attained for an **off-chip** pump

power five times less than the maximum power sent in the experiment reported here, i.e., ≈ 10 mW. We outline selected experiments that can be carried out in this parameter regime:

- Microwave-optical photon pair generation: Quantum coherent transduction is currently being pursued as a means to mediate two-qubit gates between distant qubits using optics, and correlated photon pair generation is the first step toward teleportation-based quantum state transfer. For this experiment, we will operate in the low-cooperativity regime to suppress multi-photon generation. This means it would be possible to use a pump with sub-milliwatt peak power. **Figure 1** indicates that after design adjustments, state-of-the-art pair generation rate is attainable with a CW off-chip optical pump power compatible with dilution fridge cooling power.
- Another route for improving the scalability of superconducting quantum systems is to develop “optical drives” for single qubit operations such as Pauli gates or dispersive readout, where the requisite microwave pulses are generated optically via microwave-optical transducers. Optical fibers can therefore replace most coaxial interconnects, occupying less precious fridge space and dissipating less heat in comparison. These tasks would require the generation of a substantial amount of microwave power using down-conversion and, by extension, large optical pump. As an illustration, **Figure 2** shows the maximal microwave power that can be generated on-chip through this process using a perfect unity-efficiency transducer. Assuming that the single qubit operation must be performed in ~ 1 μ s for a qubit with coherence time of ~ 100 μ s, this process requires an optical pump of at least -30 dBm. Driving other elements, such as generating the pump for quantum-limited parametric amplifiers used for readout, would require even more power and could not be achieved with less than 0 dBm optical pump. This nevertheless does not spell dead end, since unlike in the case of quantum state transfer, **the transducer can be mounted on the still (800 mK) or 3-K stage of the dilution refrigerator, which provides significantly more cooling power.**
- Finally, the heat load on the cryostat/dilution fridge can be reduced by operating the transducer in the pulsed regime with low duty cycle. This method has been employed by IST Austria for pioneering transduction experiments leveraging a peak optical pump power $\mathcal{O}(100$ mW) to enhance the $g_0 \sim 2\pi \times 10$ Hz. Our pulsed conversion demonstration serves to confirm this strategy can be used with the HBAR transducers.

Figure 2: Maximal microwave power that can be generated on-chip through down-conversion as a function of the on-chip optical pump power. The transducer is assumed to have an on-chip transduction efficiency of unity and insertion losses are neglected.

No action taken.

The authors claim that this bulk acoustic transducer design is more robust and has better power handling capabilities. This is probably true compared with other more delicate optomechanical devices. However, the bulk acoustic resonator is after all still suspended; I don't see how it can be better than integrated electro-optic transducers which have no suspended mechanical structures at all.

The HBAR transducers survived gentle press with metallic tweezers, similar to what air-clad electro-optic transducer could endure. The power handling advantage of our design is centered on the absence of parasitic effects, commonly found in electro-optic materials such as lithium niobate. For example, photorefractive effects lead to space-charge electric field from the photovoltaic drift current, which induces optical cavity resonance frequency drift via electro-optic effect. Moreover, microwave superconducting resonators are typically employed for electro-optic transducer. As a result, aside from thermal effects that lead to quality factor degradation, quasi-particle formation due to stray light also limits the pumping power and hence the maximal efficiency and microwave power generated. In our experiment, 23 dBm of CW light, limited by the light source available, engenders no sign of thermal damage.

A discussion has been added to the "Physics and design" section to clarify the point on power handling.

Another selling point highlighted by the authors is that this transducer doesn't rely on superconducting resonators. However, I don't think utilizing superconducting materials in quantum transducers is a problem or bottleneck at all because microwave applications have to be done at millikelvin temperatures anyway. Right now, the more urgent tasks are to improve the transducer efficiency and reduce the added noise. In my opinion, sacrificing the transducer performance to eliminate superconducting materials for easier fabrication or larger scale production is a step back rather than an advancement in the field.

To expand on our previous response regarding power handling (especially important for microwave power generation and qubit driving applications), we would like to emphasize that the advantage of a superconductor-less design does not lie only in room-temperature operation or large-scale manufacturing, but rather in the mitigation of sensitivity to stray light. The pump photons can be scattered off the desired path defined by the optical waveguide, at, for instance, the fiber-chip interface. Photons that end up impinging on superconducting components can break Cooper pairs, resulting in additional microwave losses. Dedicated mitigation measures are therefore required, reported to be challenging in e.g., Y. Xu et al., M. Xu et al., J. Holzgrafe et al., and T.P. McKenna et al. These manuscripts suggest the value of alternative approaches.

[3] J. Holzgrafe, N. Sinclair, D. Zhu, A. Shams-Ansari, M. Colangelo, Y. Hu, M. Zhang, K. K. Berggren, and M. Lončar, Cavity Electro-Optics in Thin-Film Lithium Niobate for Efficient Microwave-to-Optical Transduction, *Optica* 7, 1714 (2020).

[5] T. P. McKenna et al., Cryogenic Microwave-to-Optical Conversion Using a Triply Resonant Lithium-Niobate-on-Sapphire Transducer, *Optica* 7, 1737 (2020).

[6] Y. Xu, A. A. Sayem, L. Fan, C.-L. Zou, S. Wang, R. Cheng, W. Fu, L. Yang, M. Xu, and H. X. Tang, Bidirectional Interconversion of Microwave and Light with Thin-Film Lithium Niobate, *Nat Commun* 12, 4453 (2021).

Y. Xu, W. Fu, Y. Zhou, M. Xu, M. Shen, A. A. Sayem, and H. X. Tang, Light-Induced Dynamic Frequency Shifting of Microwave Photons in a Superconducting Electro-Optic Converter, *Phys. Rev. Appl.* 18, 064045 (2022).

M. Xu, C. Li, Y. Xu, and H. X. Tang, Light-Induced Microwave Noise in Superconducting Microwave-Optical Transducers, arXiv:2311.08518.

A discussion has been added to the “Physics and design” section to explicate the point on power handling.

In addition, I have some minor technical comments:

1. It would be helpful to provide all the key device parameters in the main text, such as the bulk acoustic resonator thickness, optical and acoustic Q factors, cooperativity, intra-cavity pump photon number, etc. Some of these are only shown in the supplementary material or captions of figures.

We thank the reviewer for the suggestion. The transparency of our results and ease of understanding for the reader are important to us.

A parameter summary table has been added to the main text.

2. It took me some time to understand the device structure. The cartoon in Fig. 1(c) and device photos in Fig. 2(a, b) don't really match. Maybe it will help if the authors can indicate the optical rings and coupling waveguide using some dashed lines on top of the Fig. 2(a) photo?

We thank the reviewer for the suggestion.

The optical waveguides and resonators are now indicated with dashed lines on Fig. 2 (a). Figure 1c, the device schematic, has been modified to show a different perspective to facilitate better understanding of the transduction scheme. Figures. 1d, 2d and Supplementary Figs. 3 and 7 have been modified for color scheme consistency.

3. It seems that the acoustic Q factor at room temperature is only about 267. What's the limiting factor? Is it limited by radiation loss or material loss? How high acoustic Q will the authors expect at low temperatures?

Simulations and preliminary data down to 4 K suggest the acoustic Q is currently limited by acoustic radiation loss. Offsetting the actuator from the waveguide or thinning down the waveguide constitute promising mitigation measures. **Figure 3** of this reply shows an electrode configuration that would reduce the acoustic loss without reducing the single photon coupling rate. This configuration would give a radiation loss-limited intrinsic quality factor around 5300.

No action taken.

4. Line 136-137, I think there is a typo in the superscript of η . It should be “fiber-fiber” instead of “fiber-chip” since the text says it’s fiber-to-fiber coupling efficiency?

We thank the reviewer for carefully reading our manuscript.

The typo has been corrected.

Figure 3: Alternative electrode configuration for reducing the HBAR acoustic radiation loss. (a) Cross-section of the device with stress pattern obtained from finite element simulations. (b) Corresponding microwave reflection, assuming 50 Ohms input and output ports.

Reviewer #2:

The paper presents and characterizes a novel design for a bidirectional microwave-optical converter based on a piezo-optomechanical platform. The performance of the device is comparable to that of other platforms which recently appeared in the literature. The idea of using piezoelectrical materials is not novel in the literature, and recent experiments have shown the associated potential. Nonetheless the present paper has the merit to present a novel 2D design which is able to integrate wafer-scale, CMOS-compatible HBAR and Si₃N₄ photonics technologies. A further novelty of the system is its operation in a triply resonant condition, where the mechanical mode is resonant with the splitting between the two coupled optical modes. This allows to operate with very low input power, reduce photothermal effects and open interesting possibilities for quantum operation even at room temperatures.

For this reason I think that the paper is suitable for publication Nature Communications.

We thank the reviewer for the positive assessment of our manuscript.

However I think that there is room for improvement of the presentation. In fact

1. The author could be more explicitly compare the advantages and the limitation of the proposed device with the converters recently appeared in the literature.

In **Figure 1** of this response, a detailed comparison of integrated transducers is presented, with the maximal heralding rate serving as a metric for quantum applications and microwave power generated on-chip through down-conversion for classical applications. According to these two metrics, the most competitive types of integrated transducers to date are the piezo-optomechanical crystals (OMC) and electro-optic transducers (EO). The advantages of the transducers presented here (HBAR) are the followings:

- Enhanced optical power handling capabilities: HBAR exhibits superior performance in managing optical power compared to both OMC and EO. This is evidenced by its ability to mitigate thermal effects in OMC, as well as to avoid the long-term stability issues caused by the photorefractive effect in EO.
- Improved phonon injection efficiency: HBAR surpasses OMC in phonon injection efficiency, negating the necessity for introducing additional microwave resonators. This eliminates the complexities associated with matching multiple cooperativity parameters and is expected to have a positive impact on noise figures. In the case of HBAR, internal efficiency is dictated by the optomechanical cooperativity.
- Multiplexing potential using multiple overtones: HBAR offers the unique advantage of leveraging multiple overtones for multiplexing purposes, a capability that proves challenging with OMC and EO technologies.
- Additional considerations: Other factors such as footprint and compatibility with CMOS fabrication processes also favor HBAR over its counterparts.

We have included a paragraph in the “Physics and design” subsection and adapted the introduction to elucidate the design choices we made and to draw comparisons with the limitations of piezo-optomechanical crystals and electro-optic transducers.

2. For completeness the authors could add the citation to the first theoretical proposals of mechanically mediated optical-microwave converters, which discuss both antiStokes (Y-D. Wang and A. A. Clerk, Phys. Rev. Lett. 108 153603 (2012); L. Tian, Phys. Rev. Lett. 108 153604 (2012)) and Stokes operation (S. Barzanjeh, M. Abdi, G. J. Milburn, P. Tombesi, and D. Vitali, Phys. Rev. Lett. 109, 130503 (2012).) and showed the possibility to operate these devices also in the quantum regime.

We thank the reviewer for the suggestion.

The relevant references have been added to the main text.

Reviewer #3:

The authors present an integrated optomechanical circuit that can perform frequency conversion between the microwave and optical domains, with similar efficiency when converting in each direction (i.e., microwave-to-optical and optical-to-microwave). The converter uses a combination of high-overtone bulk acoustic waves and coupled photonic ring resonators for the transduction, which is a novel design that offers several advantages. These include high coupling efficiency between the generated acoustic waves and the ring resonators through the photoelastic effect, and the ability to transmit both the optical pump and generated optical sideband frequency simultaneously and with high efficiency through the photonic integrated circuit. The primary conversion results shown in Fig. 3 are very impressive as they show that bidirectional conversion can be achieved with nearly the same frequency response in both directions. While the presented efficiency is not currently approaching the best reported to date, the novelty of the device, the bidirectional experimental results, the pulsed conversion results that point to operation at low temperature, and the high input optical power handling with linear behavior (i.e., up to 21 dBm, 126 mW) are all strong contributions. Microwave-to-optical frequency conversion is currently of significant interest to the quantum information science and cavity optomechanics communities and this paper is an important contribution to this quickly evolving field. As a result, I recommend this manuscript for publication.

We thank the reviewer for the positive assessment of our manuscript.

I've suggested a few areas below where the authors should expand explanations to improve the reader's understanding.

- What is the electrical impedance of the high-overtone bulk acoustic resonators (HBAR) and how does it affect the transduction efficiency? The planar area of the HBAR seems small compared to other HBARS that are 50 ohms at 3.5 GHz so it would be interesting to know whether the resonator is overcoupled to the microwave source (undercoupled to a sink) and whether this has a positive or negative impact on the transduction results.

The HBARS here are undercoupled to the microwave source. The admittance at the 3.480 GHz mechanical resonance is $5.5 \times 10^{-3} S$ and the admittance at the 3.165 GHz mechanical resonance is $2.5 \times 10^{-3} S$. The static capacitance of the device is around 200 fF (cf. T. Blesin et al.). The electromechanical coupling rate is determined by the overlap of the acoustic mode with the piezoelectric thin film. In particular, the number of charges in the piezoelectric layer for a given mode displacement—related to the admittance—is important. Two HBAR devices with different planar area can have the same admittance peak

Figure 4: Admittance from the microwave reflection spectrum of the HBAR. A modified Butterworth-Van Dyke model is used to fit the measurement.

and coupling rate if the smaller one induces higher charge density per vacuum displacement. **Suspending the actuator** serves to enhance the strain in the aluminum nitride film and hence admittance near resonance. As a result, the transducer presented here are relatively well coupled to the microwave port despite its small planar area. Notably, smaller planar area is preferred, since the device static capacitance contributes to microwave insertion loss. Finally, it would be desirable to overcouple the transducer to the microwave port to maximize the transduction efficiency. This can be achieved by slight adjustments in the HBAR design in order to reduce the acoustic radiation loss or increase the strain the piezoelectric film.

T. Blésin, H. Tian, S. A. Bhave, and T. J. Kippenberg, Quantum Coherent Microwave-Optical Transduction Using High-Overtone Bulk Acoustic Resonances, *Phys. Rev. A* 104, 052601 (2021).

A parameter summary table has been added at the end of the main text. The table contains the electromechanical extraction efficiency, showing the device is currently undercoupled to the source. A paragraph on the extraction of the admittance from the microwave reflection has been added in appendix.

- The modes of the coupled ring resonators shown in Fig. 2c are offset by only 3.5 GHz to match with the HBAR mode at that frequency. How was this small offset achieved? I would think that achieving this offset through lithography in a repeatable way would be very challenging. Is one resonator tuned thermally relative to the other? Are chips selected based on their mode split and then tuned? Please explain in the paper.

We thank the reviewer for prompting us to clarify the optical design. The device that was used for the main figures of the manuscript has a gap between micro-ring resonators of 800 nm. We used DUV lithography—available also in standard CMOS foundries—to ensure that the process was sufficiently precise and reproducible. The intrinsic splitting (depending only on the ring-ring gap) of the photonic molecules shows little variation over the whole wafer. The resonance frequency variation between adjacent rings can then be compensated by applying a DC bias to a piezoelectric actuator.

The design of the photonic molecule is now discussed in the main text (“Physics and design” subsection).

- Related to the previous question, piezoelectric tuning of the resonances in the coupled ring resonators is shown in Fig. 2c and thermal tuning is described in the Supplementary Information. It's not clear which method was used for collecting the data in Figs. 3 and 4 and whether one method can achieve better results if low temperature operation is not a concern. Please make this clearer in the body of the paper.

We thank the reviewer for the comment. Thermal tuning excels in applications that require low voltage levels or large tuning range; some examples include efficient single-sideband generation and polarization conversion. Piezoelectric tuning is preferred where heat load constraint becomes important, such as for cryogenic operation. The room-temperature data shown in Figs. 3 and 4 were collected using

thermal tuning. However, we checked experimentally that piezoelectric tuning gives the same transduction efficiency.

The use of thermal and piezoelectric tuning is now clarified in the main text (“Device characterization” subsection).

- In the Discussion section, the third paragraph would make more sense as the first paragraph. The third paragraph describes how the transducer could be improved through improved design and fabrication. The first and second paragraphs offer more speculative thoughts on how the transducer might be used for quantum control and networking, which while interesting, would be better after the more substantive future improvements to the transducer.

We thank the reviewer for the suggestion.

The third paragraph of the “Discussion” has been moved to be the beginning of this section.

Summary of changes to the manuscript

- The g_0 improvement is now explicitly mentioned in the conclusion.
- Parameter summary table has been added to the main text.
- Figure 2a now shows the optical waveguides in red.
- Figure 1c has been modified to show a different view on the device schematic. Figures 1d, 2d and Supplementary Figs. 3 and 7 have been modified to maintain a consistent color scheme.
- The typo on line 137 has been corrected.
- A paragraph has been added in the “Physics and design” section to clarify the design choices, making comparison with the existing state-of-the-art more explicit.
- References on the Stokes and Anti-Stokes processes for microwave-optical conversion have been added in the “Physics and design” section.
- The extraction of the admittance and the role of the cladding suspension in electromechanical coupling are now explained in the appendix.
- The design of the photonic molecule is now discussed in the main text.
- The use of thermal and piezoelectric tuning is now clarified in the main text.
- The third paragraph of the “Discussion” section has been moved to be the first.
- The appendix on the fabrication process flow is now complemented by Supplementary Fig. 7.
- The bibliography has been updated to reflect the publication of manuscripts posted as preprints at the time of the original submission (references 22 and 23).
- Minor stylistic edits (consistency of the name for g_0 , efficiencies appearing in dB, wording for scandium doping ...).
- A data availability statement has been added.

REVIEWERS' COMMENTS

Reviewer #2 (Remarks to the Author):

The new version has significantly improved the presentation of the results. The advantages of the demonstrated platform are now much better clarified. Also the novel features of the design are much better clarified and strengths and weaknesses (together with the possibility for improvements) are more clearly described. In my opinion the paper is suitable for publication in its present form.

Reviewer #3 (Remarks to the Author):

The authors have addressed my feedback and I'm satisfied with their changes. In general, the paper is significantly improved, with additional detail on the optomechanical design and a more complete performance comparison with the literature. While the presented device does not match the best performance reported to date for microwave-to-optical frequency conversion, the converter design is novel and the results point to strong prospects for improved performance in the future. As a result, I believe that it is appropriate for publication in its current form.

**Response to referees for the manuscript
“Bidirectional microwave-optical transduction based on
integration of high-overtone bulk acoustic resonators and
photonic circuits”**

We appreciate the careful reading of our manuscript by the reviewers and their constructive comments; their suggestions have certainly helped improve the quality of the manuscript.

We present a point-by-point response to each of the reviewers' comments. The reviewers' original suggestions are in black, our replies in blue.

Reviewer #2:

The new version has significantly improved the presentation of the results. The advantages of the demonstrated platform are now much better clarified. Also the novel features of the design are much better clarified and strengths and weaknesses (together with the possibility for improvements) are more clearly described. In my opinion the paper is suitable for publication in its present form.

We thank the reviewer for their constructive comments and the support of our manuscript for publication.

Reviewer #3:

The authors have addressed my feedback and I'm satisfied with their changes. In general, the paper is significantly improved, with additional detail on the optomechanical design and a more complete performance comparison with the literature. While the presented device does not match the best performance reported to date for microwave-to-optical frequency conversion, the converter design is novel and the results point to strong prospects for improved performance in the future. As a result, I believe that it is appropriate for publication in its current form.

We thank the reviewer for their valuable feedback and for recognizing the novelty of our design.

Summary of changes to the manuscript

- The abstract has been slightly shortened to fit the 150-word limit described in the Nature Communications formatting instructions.
- A Methods section has been added to comply with the author checklist instructions.
- A competing interests statement has been added.
- The article structure now follows the author checklist instructions. In particular, the acknowledgements and author contributions have been moved after the references, and Table I now appears before Fig. 1.
- The color highlighting indicating the last modifications has been removed.
- The font of subscripts and superscripts for abbreviations has been change from italic to roman (e.g., ω_m became ω_{m}).
- The phrasing of the data availability statement has been changed to match the provided template.
- Minor grammar and spelling mistakes have been corrected.
- The position of the figures in the Supplementary Information has been changed. They are now placed within the text instead of at the end of the document.